# Gas-Phase Fluorination of g-C_3_N_4_ for Enhanced Photocatalytic Hydrogen Evolution

**DOI:** 10.3390/nano12010037

**Published:** 2021-12-23

**Authors:** Lidong Sun, Yu Li, Wei Feng

**Affiliations:** 1School of Materials Science and Engineering, Tianjin University, Tianjin 300072, China; sunlidong@tju.edu.cn; 2Key Laboratory of Advanced Ceramics and Machining Technology Ministry of Education, Tianjin 300072, China

**Keywords:** fluorinated g-C_3_N_4_, fluorination, photocatalytic

## Abstract

Graphitic carbon nitride (g-C_3_N_4_) has attracted much attention because of its potential for application in solar energy conservation. However, the photocatalytic activity of g-C_3_N_4_ is limited by the rapidly photogenerated carrier recombination and insufficient solar adsorption. Herein, fluorinated g-C_3_N_4_ (F-g-CN) nanosheets are synthesized through the reaction with F_2_/N_2_ mixed gas directly. The structural characterizations and theoretical calculations reveal that fluorination introduces N vacancy defects, structural distortion and covalent C-F bonds in the interstitial space simultaneously, which lead to mesopore formation, vacancy generation and electronic structure modification. Therefore, the photocatalytic activity of F-g-CN for H_2_ evolution under visible irradiation is 11.6 times higher than that of pristine g-C_3_N_4_ because of the enlarged specific area, enhanced light harvesting and accelerated photogenerated charge separation after fluorination. These results show that direct treatment with F_2_ gas is a feasible and promising strategy for modulating the texture and configuration of g-C_3_N_4_-based semiconductors to drastically enhance the photocatalytic H_2_ evolution process.

## 1. Introduction

The drastically increasing energy consumption globally requires clean and sustainable energy as alternatives to fossil fuels. Hydrogen energy is considered as the most promising candidate due to its potential application in fuel cells without carbon emissions. As such, the hydrogen evolution process has been widely studied, involving water splitting driven by light and with the help of a photocatalyst [1,2,3]. Graphitic carbon nitride (g-C_3_N_4_) has been demonstrated as a promising solar-to-fuel, metal-free, polymeric photocatalyst due to its suitable band structure, highly thermal and chemical stability, low cost and convenient preparation [4,5,6]. However, there are some intrinsic drawbacks to g-C_3_N_4_, such as its low electrical conductivity, insufficient light absorption, poor surface area and rapid recombination of photogenerated charge carriers [7,8]. Therefore, a great many strategies have been explored to improve its photocatalytic performance, including morphology design, elemental doping, heterojunction construction, nanocomposite hybridization and photosensitizer decoration [9,10,11]. Among these various strategies, it has been acknowledged that the introduction of non-metal heteroatoms is an effective method for enhancing the photocatalytic activity of semiconductors by expanding the absorption range of the solar spectrum, as highlighted in the pioneering study by Asahi et al. [12]. So far, various non-metal heteroatoms have been applied to synthesize doped g-C_3_N_4_ with significantly improved photocatalytic activity because of the narrowed bandgap, extended visible light absorption and promoted charge separation [13,14,15,16]. Fluorine (F) is the most electronegative element in the periodic table and the C-F bond is the strongest single covalent bond, meaning fluorination could provide maximum charge polarization to host materials, enhance the activity of energy-related reactions and improve the stability of compounds [17]. Nowadays, F-doped, nanostructured carbons have attracted tremendous interest and obtained great success, having controllable mechanical, electronic and magnetic properties [18,19,20]. F-doped g-C_3_N_4_ can be synthesized by either heating a mixture of dicyandiamide as the precursor with inorganic fluorides at high temperature [21,22,23] or via post-fluorination of g-C_3_N_4_ with the F-containing solution hydrothermally [24,25,26,27,28,29]. Both experimental and theoretical investigations have revealed that the structural distortion induced by F-doping enhances the photocatalytic activity of g-C_3_N_4_ because of the narrowed bandgap and accelerated separation of photogenerated electron–hole pairs. However, F-doped g-C_3_N_4_ is currently achieved through the nucleophilic substitution reaction of F with positively charged carbon atoms in a g-C_3_N_4_ framework [30], which not only limits the doping level but also restricts the degree of structural modification. Therefore, it is desirable to explore novel F-doping routes to increase the doping level and to modulate the structure to improve the photocatalytic properties of g-C_3_N_4_ further.

In addition to the hydrothermal method involving the use of F^-^ as the elemental source, treatment with F-containing gas is another widely applied method to introduce F into the carbonaceous basal plane, with the molar ratio of F to C (F/C) in the range of 0–1.12 [31], which is called the gas-phase fluorination process [32,33]. Based on the consideration that bulk g-C_3_N_4_ has a multi-layer structure similar to graphite [34], as well as the successful fluorination of graphite through the direct gas fluorination method [35], it can be expected that the reaction with F_2_ is likely to introduce F atoms into the g-C_3_N_4_ framework at relatively high levels. However, the existence of defect-rich, N-bridged “poly(tri-s-triazine)/poly(triazine)” derived from the incomplete de-amination or polymerization of the nitrogen-containing precursor in g-C_3_N_4_ [36], rather than achieving a perfectly conjugated structure such as in graphite, should endow g-C_3_N_4_ a different fluorination mechanism [37]. Therefore, the morphological and electronic changes in g-C_3_N_4_ with F_2_ gas are worth investigating.

In this work, we prepared fluorinated g-C_3_N_4_ (F-g-CN) using F_2_ gas at high temperatures for the first time. Unlike conventional reactions with F^-^, the extremely strong oxidizability of F_2_ gas introduces F atoms into the g-C_3_N_4_ framework at a relatively high level through the addition reaction, leads to drastic structural distortion due to the transformed hybridization mode from sp^2^ to sp^3^ and creates abundant nitrogen defects by etching bridged N atoms simultaneously, thereby improving the charge separation because of the affected π band and long-pair electron states. As a result, F-g-CN shows 11.6 times greater photocatalytic activity for hydrogen production under visible light compared to pristine g-C_3_N_4_ thanks to the increased surface area of the porous structure, enhanced light absorption and efficient charge separation. This study reveals the effectiveness of F_2_ gas in improving the photocatalytic activity of g-C_3_N_4_ because of the synergetic effects of nitrogen defects, structural distortion and fluorine species.

## 2. Materials and Methods

### 2.1. Preparation of the Samples

The pristine g-C_3_N_4_ was prepared by calcining melamine (3 g) at 550 °C in argon for 4 h with a heating rate of 2.3 °C/min in a tube furnace. F-g-CN was synthesized under a static fluorination atmosphere in a nickel reactor. At first, 0.3 g of synthesized g-C_3_N_4_ power was placed on a nickel boat and then transferred into the nickel reactor. After sealing up the reactor carefully, the air and moisture were excluded from this reactor by alternately evacuating and filling the reactor with N_2_ gas three times. Subsequently, mixed gas of F_2_/N_2_ (20 vol.% of F_2_) was entered into the reactor slowly until 0.1 MPa at room temperature. Then, the reactor was heated at a rate of 5 °C min^−1^ until reaching the target temperature and kept for 1 h to ensure sufficient fluorination. The schematic synthesis process of F-g-CN is illustrated in Figure 1. After the reaction, the residual F_2_ and byproducts were ejected from the reactor by N_2_ gas and absorbed on an alkali absorbent. Finally, F-g-CN powder was obtained after naturally cooling to room temperature.

### 2.2. Characterization

The morphology of the samples was characterized by field emission scanning electron microscopy (SEM, Hitachi S4800, Tokyo, Japan) and transmission electron microscopy (TEM, JEM2100F, Joel, Tokyo, Japan). Moreover, the energy-dispersive spectroscopy (EDS) profiles were recorded. The crystal structures were analyzed by X-ray diffraction (XRD, Bruker D8 advanced with Cu Kα radiation). The specific surface area and pore size distribution were obtained using nitrogen adsorption–desorption at 77 K on an Autosorb-iQ2-MP (Quantachrome Inst., Boynton Beach, FL, USA). Fourier transform infrared spectra (FTIR) were recorded on a Bruker tensor 27 FTIR spectrometer ranging from 4000 to 400 cm^−1^. X-ray photoelectron spectroscopy (XPS) results were measured in Thermo ESCALAB 250 with Al Kα X-rays. UV–Vis diffuse reflectance spectra (DRS) were performed with a UV-Vis spectrophotometer (Perkin Elmer Lambda750) using BaSO_4_ as the reflectance standard reference. Steady-state photoluminescence (PL) spectra were recorded on a spectrophotometer (Hitachi F-4600, Hitachi, Japan). The time-resolved spectra were detected using a fluorescence spectrometer (Edinburgh FLS980, Instruments Ltd., UK) under an excitation wavelength of 360 nm at room temperature.

### 2.3. Photoelectrochemical Measurement

The photoelectrochemical measurements were conducted on an electrochemical station (CHI660D, Shanghai Chenhua Limited, Shanghai, China) with a standard three-electrode system in 0.5 M Na_2_SO_4_ solution. A Pt foil was used as the counter electrode and Ag/AgCl (saturated KCl) was used as the reference electrode. The working electrode was fabricated by coating the slurry of the 5 mg photocatalyst dispersed in 0.5 mL ethanol and 20 μL nafion on a clean fluoride-doped tin oxide (FTO) glass. After being naturally dried in air, the electrode was tested under the light source of a 300 W Xe lamp with a filter (λ > 420 nm). The photocurrent responses of the photocatalyst were recorded using a 20 s on–off light cycle at a voltage of 0.0 V. Electrochemical impedance spectroscopy (EIS) results were measured in the frequency range of 0.05 Hz to 100 kHz with a perturbation of 5 mV.

### 2.4. Photocatalytic Measurement

The photocatalytic hydrogen production experiments of samples were evaluated in an online photocatalytic hydrogen production system (CEL-SPH2N, Ceaulight, Beijing, China) with a 420 nm cutoff filter. Typically, 50 mg of photocatalytic powder was suspended in an aqueous solution (100 mL) containing triethanolamine (10 vol.%) as a sacrificial electron donor. In addition, 3 wt.% Pt was introduced as co-catalyst via in situ photodeposition of H_2_PtCl_6_. The reaction solution was evacuated several times to remove air completely prior to radiation under visible light. The temperature of the reaction was maintained at 25 °C during the reaction. The produced hydrogen was measured with an online gas chromatograph (GC-14C, Shimadzu, Kyoto, Japan) equipped with a thermal conductive detector (TCD) using nitrogen as a carrier gas. The apparent quantum efficiency (AQE) for H_2_ evolution was measured under identical experimental conditions, except that monochromatic light was used for irradiation with a 420 nm band pass filter. The AQE was calculated based on Equation (1):(1)$$\mathrm{AQE}\;=\;\frac{{2\;\times \;\mathrm{number}\;\mathrm{of}\;\mathrm{evolved}\;H}_{2}}{\mathrm{number}\;\mathrm{of}\;\mathrm{incident}\;\mathrm{photos}\;}\;\times \;100\%$$

### 2.5. Theoretical Calculations

First-principle calculations within the framework of density functional theory (DFT) were performed with the DMol^3^ software package. Perdew–Burke–Ernzerh (PBE) exchange correlation functionals with general gradient approximation (GGA) were utilized for geometric optimization and single-point energy calculations. Effective core potentials with a double numerical basis were employed for the description of core electrons. In the structural optimization calculations, the convergence tolerances for energy, maximum force and maximum displacement were set to 1.0 × 10^−6^ Ha per atom, 1.0 × 10^−4^ Ha/Å and 1.0 × 10^−3^ Å, respectively. The energy tolerance for self-consistent field iterations in the energy calculations was set to 1.0 × 10^−6^ Ha. The Brillouin zone was sampled at the 5 K × 5 K × 3 K point, because there was no significant change in the calculated energies when a larger number of K points were applied. DFT-D corrections were carried out using the Grimme method for all calculations.

## 3. Results and Discussion

### 3.1. Morphology and Structural Analysis

The morphologies of g-C_3_N_4_ and prepared F-g-CN at different temperatures were examined by SEM at first. From the SEM image of g-C_3_N_4_ in Figure 2a, the dense stacked lamellar structure can be observed, which suppresses the capability of light absorption as well as the charge and mass transport [38]. The fluorination temperature is a crucial parameter in controlling the morphology and the fluorine content of g-C_3_N_4_; hence, g-C_3_N_4_ was fluorinated at 120, 150 and 180 °C, labeled as F-g-CN-120, F-g-CN-150 and F-g-CN-180, respectively, to investigate the photocatalytic activity. After the fluorination, all F-g-CN samples changed to porous and loose particles (Figure 2b and Appendix A). The corresponding TEM image of g-C_3_N_4_ shown in Figure 2c further confirms that g-C_3_N_4_ is constructed of large and overlapped nanosheets. From the TEM images of F-g-CN in Figure 2d and Appendix A, it can be noticed that the size of these layers is decreased remarkably and a large number of pores can be noticed clearly, consistent with its SEM image. In addition, these pores become larger with the increase in fluorination temperature, and the porous structure of F-g-CN obtained at 180 °C is seriously deteriorated due to the extensive fluorination at high temperatures. It should be noted that the much more transparent TEM image of F-g-CN than that of g-C_3_N_4_ indicates the reduced thickness by fluorination. The extremely reactivity of F_2_ gas destroys the structure of g-C_3_N_4_, leading to lots of pores in the g-C_3_N_4_ structure, similar to the porous g-C_3_N_4_ synthesized by strong oxidants [39]. Furthermore, the extremely low yield of F-g-CN fluorinated at 180 °C is insufficient for photocatalytic needs. Thus, F-g-CN fluorinated at 150 °C was selected for further investigation. The F-g-CN that appears in the following text refers to F-g-CN-150 if no special description is given.

Based on N_2_ adsorption–desorption measurements at 77 K for g-C_3_N_4_ and F-g-CN, both of them exhibit type IV isotherms with H3 hysteresis loops (Appendix A), indicating the mesoporous structure of these samples [40]. Compared with g-C_3_N_4_, the shifted hysteresis loop of F-g-CN to the lower relative pressure region and the enlarged area of the hysteresis loop demonstrate the increased mesoporous structure after fluorination, in agreement with the morphological results. The specific surface areas calculated by the Brunauer–Emmett–Teller (BET) method for g-C_3_N_4_ and F-g-CN were 11.5 and 74.0 m^2^ g^−1^, respectively. The greatly increased surface area after fluorination can be ascribed to the generation of abundant pores. In addition, the pore size distributions of g-C_3_N_4_ and F-g-CN as analyzed using the Barrett–Joyner–Halenda (BJH) method further confirmed the plentiful mesopores in the range of 20–100 nm after the fluorination process (Appendix A). Therefore, F_2_ gas has a great impact on the formation of the porous structure in g-C_3_N_4_.

The XRD patterns of g-C_3_N_4_ and F-g-CN in Figure 3a display a similar crystalline structure. The main diffraction peaks at 13.1° and 27.3° for all the samples correspond to the in-plane repetition of the tri-s-triazine motif and the periodic stacking of conjugated aromatic structures, respectively [37]. Note that the diffraction peak at 13.1° for F-g-CN almost disappears, which demonstrates the overwhelmingly destructed in-plane structure of g-C_3_N_4_ caused by the substantial mesopores, as revealed by the size distribution analysis [41]. In addition, the intensity of the diffraction peak at 27.3° for F-g-CN is weak in comparison with g-C_3_N_4_, suggesting the disturbed periodic structure of stacked g-C_3_N_4_ planes caused by the fluorination process [13]. The FTIR spectra of these samples (Figure 3b) confirm that the molecular skeletal vibration modes of F-g-CN are close to those of g-C_3_N_4_. The peak at 810 cm^−1^ is assigned to the characteristic breathing mode of triazine ring. The peaks between 900 and 1800 cm^−1^ are ascribed to typical stretching vibration modes of C=N and C-N heterocycles. The broad peak between 3000 and 3600 cm^−1^ corresponds to N-H and O-H stretching vibration modes [42]. However, the stretching vibration of fluorinated species occurs in the range of 1000 to 1400 cm^−1^, which is overlapped with the region associated with the skeletal stretching vibrations of aromatic C-N heterocycles from 900 to 1800 cm^−1^, so it is difficult to distinguish fluorinated species from the FTIR spectrum of F-g-CN. Moreover, the absence of the peak at 2180 cm^−1^ in F-g-CN demonstrates that nitrogen defects of cyano groups are not formed after the fluorination process [43].

The XPS measurement was carried out to investigate the chemical states and compositions of g-C_3_N_4_ and F-g-CN. The XPS survey spectrum of F-g-CN directly demonstrates the successful fluorination of g-C_3_N_4_ with the F concentration of 7.05 at. % (Appendix A). In addition, the elemental mapping images of F-g-CN (Appendix A) with homogeneous distribution of C, N and F elements further demonstrates the effective fluorination of g-C_3_N_4_. By comparing the high-resolution C1s spectra of g-C_3_N_4_ and F-g-CN (Figure 4a), it can be observed that besides the three peaks presented in g-C_3_N_4_, which are assigned to the adventitious carbon species (284.8 eV), C-NH_X_ on the edges of heptazine units (286.3 eV) and N-C=N coordination in g-C_3_N_4_ framework (288.4 eV) [44,45], a new peak appears at 289.5 eV in the C1s of F-g-CN that is ascribed to C-F bond, because F atoms tend to bond with C atoms with lower electronegativity than nitrogen atoms [19]. Meanwhile, the high-resolution N1s spectra for these samples (Figure 4b) exhibit the similar features without any additional peaks, which further proves the formation of C-F bonds rather than N-F bonds in F-g-CN. The apparent three peaks at about 398.8, 399.9 and 401.2 eV correspond to sp^2^ hybridization nitrogen (C=N-C), sp^3^ hybridization nitrogen (N-C_3_) and amino functional groups (C-N-H), respectively [42,44]. The high-resolution F1s spectrum of F-g-CN (Figure 4c) can be resolved into two peaks. The peak at 685.1 eV is assigned to physically absorbed or entrapped F atoms because of the abundant formed mesopores and enlarged surface area [22], while the peak at 688.0 eV is identified as C-F bonds that have been revealed in its C1s spectrum [46]. The binding energies of C1s, N1s and F1s spectra of g-C_3_N_4_ and F-g-CN are listed in Appendix A, which shows some variation after fluorination. However, the intensity ratio of N-sp^2^ to N-sp^3^ decreases from 3.34 to 1.32 after fluorination, which indicates the intensive structural distortion of the original conjugated g-C_3_N_4_ planes caused by formed C-F bonds [27].

To obtain more information about g-C_3_N_4_ structural deformation during fluorination, the first-principle DFT calculations were performed. The g-C_3_N_4_ crystal cell (7.12 Å × 7.12 Å × 9.30 Å) was modeled and optimized (Appendix A). As mentioned in the discussion of the C1s spectrum of F-g-CN, F_2_ molecules tend to react with C atoms rather than N atoms. Taking this into consideration, the two nonequivalent carbon atoms in bulk g-C_3_N_4_ are designated C1 and C2 in Appendix A. To determine the active site for fluorination, the Mulliken charge population analysis is applied to the bulk g-C_3_N_4_ model [47,48]. The C1 atom bound to the tri-coordinated bridged N atom has the higher Mulliken charge value, so these positively charged C atoms associate more readily with the F-bearing intermediate. Therefore, the F-g-CN model is built by introducing one F atom bonded with one C1 atom into each layer of g-C_3_N_4_. The F proportion obtained with this model is 6.67 at. %, which closely approximates the F proportion of 7.05 at. % (Appendix A) determined from the XPS spectrum. In the optimized F-g-CN structure (Appendix A), fluorine atoms cause significant deformation of the C-N plane due to the disturbance of the conjugated heptazine unit. The C-F bond distance in F-g-CN is 1.33 Å, which is similar to the covalent C-F bond length in the CF_4_ crystal [49], indicating the strong bonding interactions between F and C atoms. However, the length of the bond between C1 and the tri-coordinated bridged N atom increases drastically from 1.47 Å to 2.90 Å. This is drastically longer than the covalent C-N bond, which shows there is no short range between these two atoms. Considering that the single bond of N-C_3_ is weaker than conjugated double bonds of C=N-C in the heptazine ring, the bridging nitrogen atoms are easily removed [50] after the presence of structural distortion induced by a high content of F under fluorine atmosphere. The unpaired electrons on the tri-coordinated bridged N atom resulting from the breakage of the C-N bond make it prone to reacting with F radicals. As a result, N atoms depart from the g-C_3_N_4_ framework in the form of NF_3_ gas during fluorination, leading N-vacancy defects in these positions, simultaneously associated with the formation of covalent C-F bond. The decrease in N responsible for these defects is revealed in the XPS survey spectrum of F-g-CN. Therefore, the reaction with F_2_ gas generates N-vacancy defects and causes structural distortion of g-C_3_N_4_, as illustrated in Figure 4d.

### 3.2. Optical and Electronic Properties

The optical absorption properties of the samples were studied by UV–Vis DRS spectra (Figure 5a). The absorption edge of F-g-CN shows an obvious blue shift compared with g-C_3_N_4_, which can be also represented by the color variation from yellow bulk to light yellow after the fluorination process (insets in Figure 5a). The corresponding bandgaps of the samples are determined by Tauc plots using the Kubelka–Munk function (Figure 5b), and the bandgap increases from 2.66 eV for g-C_3_N_4_ to 2.69 eV after fluorination, which is attributed to the structural distortion of F-g-CN [27]. Generally, the enlarged bandgap suffers from the shortcoming of light absorption, although an optical absorption tail in the longer wavelength observed in F-g-CN clearly implies the generation of a defect state or heteroatom level [51,52].

The influence of fluorination on the electronic structure of g-C_3_N_4_ is further investigated by valance band (VB) XPS spectra in Figure 5c. The positions of VB maximum of g-C_3_N_4_ and F-g-CN are validated as 2.46 and 2.00 eV, respectively. In combination with the bandgap derived from UV–Vis DRS, the conduction band (CB) edges of g-C_3_N_4_ and F-g-CN are estimated to be –0.20 and –0.69 eV, respectively. The CB minimum of F-g-CN is notably more negative than that of g-C_3_N_4_, which implies the more powerful reduction ability of the former [53]. The upshift of CB after fluorination is supposed to be caused by the vigorous structural distortion that is uncovered by the structural characterizations above. The schematic energy band structures of g-C_3_N_4_ and F-g-CN are illustrated in Figure 5d.

The steady PL spectra also depict a slight blue shift of the emission peak in F-g-CN (Figure 6a), consistent with their UV–Vis DRS spectra, which are proposed represent the induced structural distortion and formed C-F bonds. The time-resolved PL spectra were carried out at the wavelength of the maximum emission peak to measure the lifetime of the photogenerated charge carriers. The time-resolved PL spectra were carried out at the wavelength of the maximum emission peak to measure the lifetime of the photogenerated charge carriers. The fluorescence decay curves (Figure 6b) were fitted by using the following equation: I(t) = A_1_exp (−t/τ_1_) + A_2_exp (−t/τ_2_) + A_3_exp (−t/τ_3_). In order to relate the overall emission decay behaviors, the average lifetime (τ_ave_) was evaluated using the following equation: τ_ave_ = (A_1_τ_1_^2^ + A_2_τ_2_^2^ + A_3_τ_3_^2^)/(A_1_τ_1_ + A_2_τ_2_ + A_3_τ_3_). It was clearly deduced that after fluorination, the average emission lifetimes of g-C_3_N_4_ and F-g-CN were 6.39 and 8.62 ns, respectively. The fluorescence lifetime of F-g-CN was longer than that of g-C_3_N_4_, suggesting that more charge carriers took part in the photocatalytic reaction [54,55].

Transient photocurrents under visible light irradiation and EIS were implemented to analyze the photogenerated charge separation and transfer performances of the prepared samples. As displayed in Figure 6c, both g-C_3_N_4_ and F-g-CN electrodes shows positive transient photocurrents via alternate on–off cycles of visible light irradiation at a constant applied potential. Obviously, the photocurrent density values of g-C_3_N_4_, F-g-CN-120, F-g-CN-150 and F-g-CN-180 are 3.54, 4.58, 8.82 and 4.28 μAcm^−2^, respectively, indicating that F-g-CN-180 can provide more photogenerated charge carriers. The photocurrent density of the F-g-CN electrode is much higher than that of the g-C_3_N_4_ electrode, confirming the improved separation efficiency of photogenerated electron–hole pairs in F-g-CN, which is modulated by the distorted structure after the fluorination [27]. The density of state (DOS) of g-C_3_N_4_ (Figure 7a) shows that its VB is dominated by ring N atoms, while its CB is composed of C atoms and a small proportion of ring and inner N atoms. It can be noted that from the DOS of F-g-CN (Figure 7b), the CB of F-g-CN is dominated by the same atoms as g-C_3_N_4_, although its VB is comprised of ring N and bridged N atoms. The bridged N atoms in the VB of F-g-CN contribute to the mobility of photogenerated electron–hole pairs due to the presence of C–F bonds and the distorted structure [10]. Moreover, the arc radius of the EIS Nyquist plot for F-g-CN is markedly smaller than that of g-C_3_N_4_ (Figure 6d), which reflects the decreased charge transfer resistance at the interface and benefits the charge transfer after fluorination [56]. Therefore, the efficiency of the charge separation is much higher in F-g-CN than pristine g-C_3_N_4_.

### 3.3. Photocatalytic Performance

The photocatalytic performances of g-C_3_N_4_ and F-g-CN samples were evaluated with hydrogen evolution under visible light illumination (λ > 420 nm), where 3 wt.% Pt and TEOA were added in an aqueous solution as a co-catalyst and sacrificial agent, respectively. Both of the samples show the H_2_ evolution activity in Figure 8a and Appendix A, while the average H_2_ evolution rates of g-C_3_N_4_, F-g-CN-120, F-g-CN-150 and F-g-CN-180 are 112, 675, 1298 and 370 μmol h^−1^ g^−1^, respectively. It can be seen that the H_2_ evolution rate of F-g-CN-150 is 11.6 times higher than that of g-C_3_N_4_ and the calculated AQE for F-g-CN-150 is as high as 4.53% at 420 nm, which highlights the outstanding superiority of F-g-CN-150 in photocatalytic H_2_ production (Appendix A). Furthermore, there is no obvious decline in photocatalytic H_2_ evolution for five cycles under the same testing conditions (Figure 8b), indicating the sufficient stability and durability of F-g-CN as the photocatalyst. Based on the characterizations of F-g-CN, it can be claimed that the fluorination by F_2_ gas is an effective method to prominently promote the photocatalytic H_2_ production. Therefore, the synergetic effects of the highly porous structure, the considerable nitrogen vacancy defects and the relatively high degree of fluorination contribute to the excellent photocatalytic H_2_ evolution activity of F-g-CN.

## 4. Conclusions

In summary, a porous structure, nitrogen vacancy defects and fluorinated species were effectively introduced into the g-C_3_N_4_ framework via direct fluorination with F_2_ gas. Compared with the pristine g-C_3_N_4_, the photocatalytic H_2_ evolution activity of F-g-CN was improved significantly with a rate of 1298 μmol h^−1^ g^−^^1^, which was 11.6 times higher than that of g-C_3_N_4_ under the same conditions. Based on the first-principle DFT calculations, detachment of bridged N atoms in tri-s-triazine from the g-C_3_N_4_ framework occurred during fluorination, resulting in N vacancy defects and covalent C-F bonds at the edge of the fragment. The specific surface area of F-g-CN was markedly enlarged due to the formation of mesopores in the g-C_3_N_4_ structure. Electron transition was also affected by the N vacancy defects, as well as by the presence of introduced F atoms and the structural distortion induced by fluorination. As a result of these synergetic effects, the separation of photogenerated charge carriers and utilization of solar energy were improved. Our study has highlighted a unique means of improving the performance of photocatalysts through fluorination. These results are instructive for understanding the complex effects of fluorination to g-C_3_N_4_, which inspired us to optimize the photocatalytic performance by engineering a photocatalytic texture and electronic structure.

## Figures and Tables

**Figure 1 nanomaterials-12-00037-f001:**
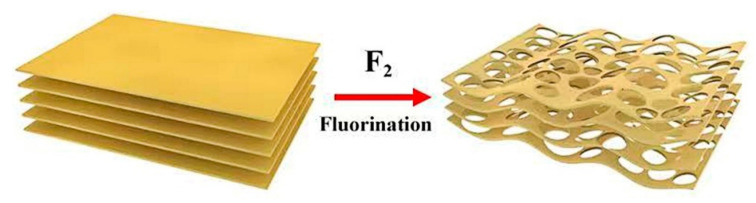
The schematic synthesis process of F-g-CN.

**Figure 2 nanomaterials-12-00037-f002:**
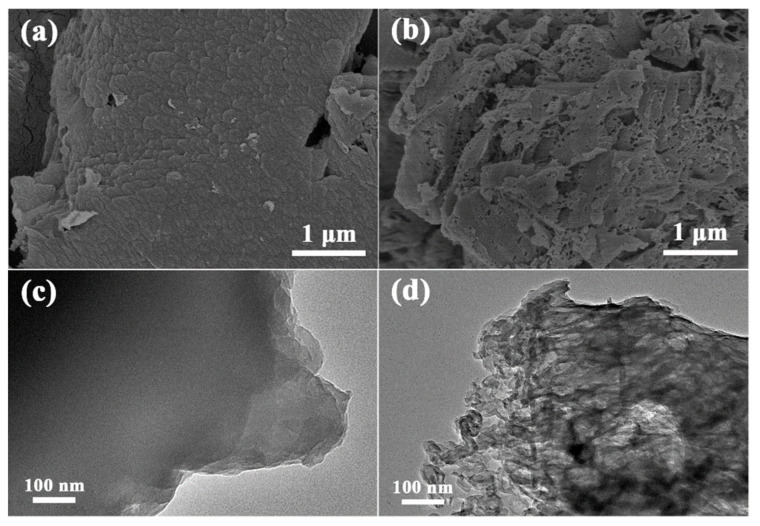
SEM and TEM images of (**a**,**c**) g-C_3_N_4_ and (**b**,**d**) F-g-CN synthesized at 150 °C.

**Figure 3 nanomaterials-12-00037-f003:**
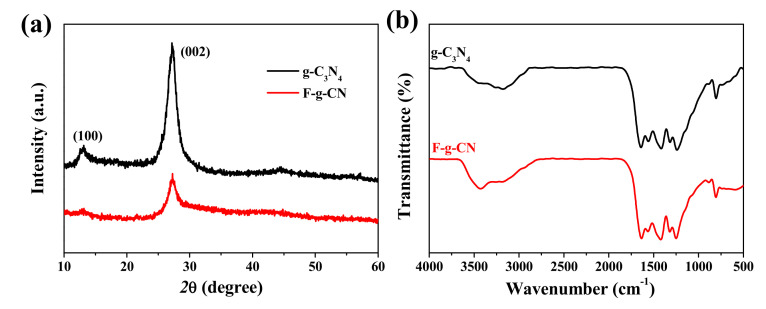
(**a**) XRD patterns and (**b**) FTIR spectra of g-C_3_N_4_ and F-g-CN.

**Figure 4 nanomaterials-12-00037-f004:**
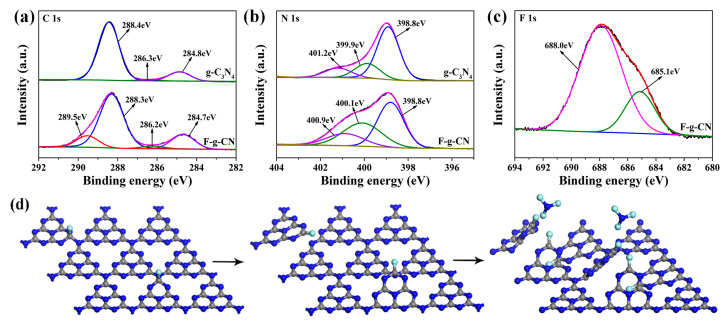
High-resolution XPS spectra of g-C_3_N_4_ and F-g-CN: (**a**) C1s; (**b**) N1s; (**c**) F1s. (**d**) Schematic diagram of the g-C_3_N_4_ fluorination process. The blue, gray and cyan circles represent nitrogen, carbon and fluorine atoms, respectively.

**Figure 5 nanomaterials-12-00037-f005:**
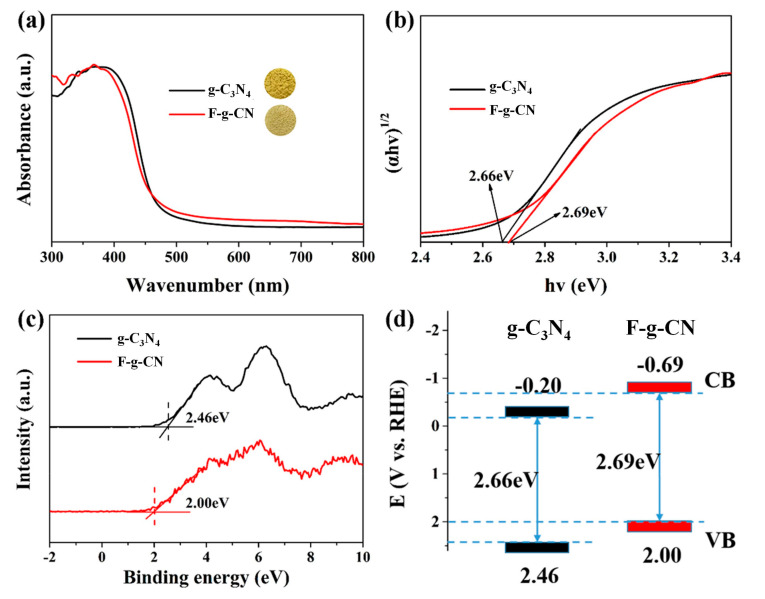
(**a**) UV-Vis absorption spectra of g-C_3_N_4_ and F-g-CN and (**b**) their corresponding bandgaps. (**c**) VB XPS spectra of g-C_3_N_4_ and F-g-CN. (**d**) Schematic illustration of the g-C_3_N_4_ and F-g-CN band structures.

**Figure 6 nanomaterials-12-00037-f006:**
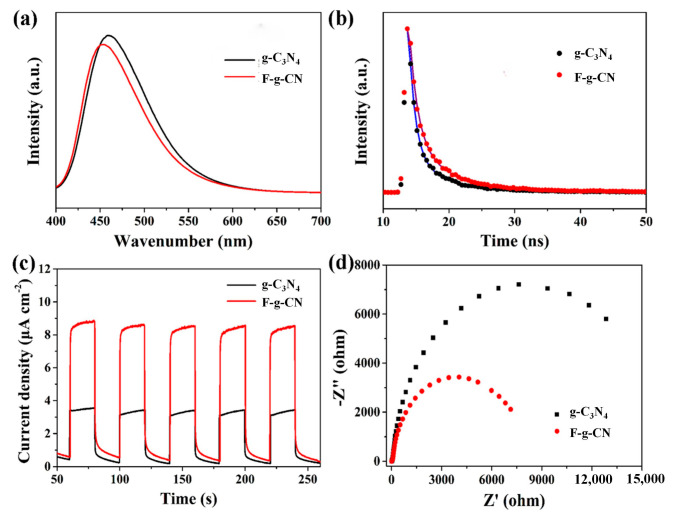
(**a**) PL spectra and (**b**) time-resolved PL spectra of g-C_3_N_4_ and F-g-CN. (**c**) Transient photocurrent responses and (**d**) EIS plots of g-C_3_N_4_ and F-g-CN.

**Figure 7 nanomaterials-12-00037-f007:**
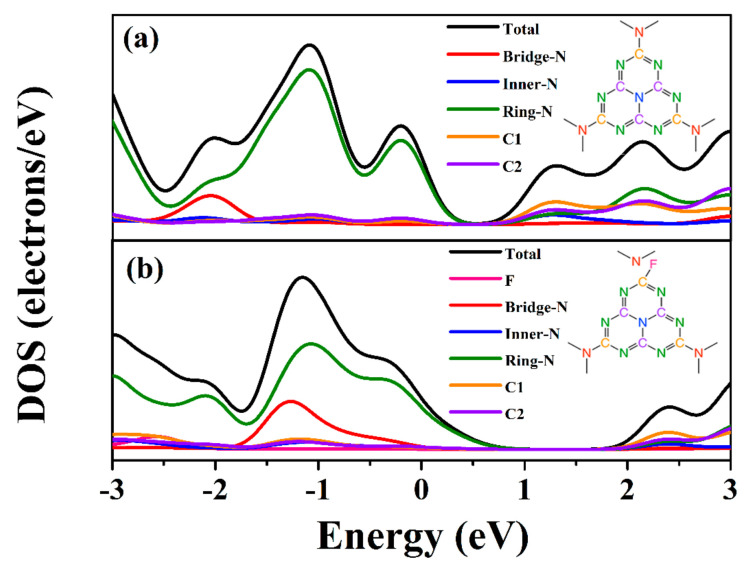
Calculated density of state (DOS) values for (**a**) g-C_3_N_4_ and (**b**) F-g-CN from their optimized structures.

**Figure 8 nanomaterials-12-00037-f008:**
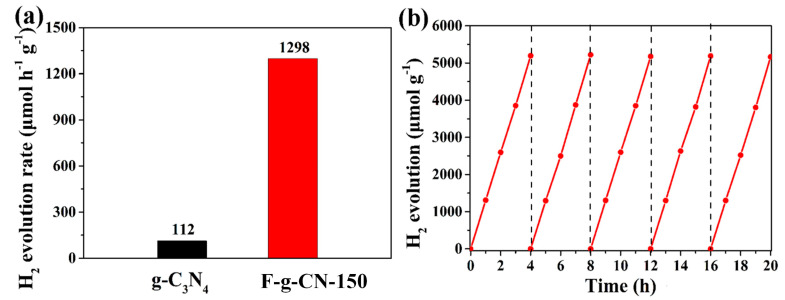
(**a**) Photocatalytic H_2_ evolution rates for g-C_3_N_4_ and F-g-CN. (**b**) Cycling tests of H_2_ evolution with F-g-CN-150.

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
