# Peer review of "Gas-Phase Fluorination of g-C3N4 for Enhanced Photocatalytic Hydrogen Evolution"

_nanomaterials, 2021, doi:10.3390/nano12010037_

Round 1

Reviewer 1 Report

This manuscript is a very interesting work on the synthesis of a material based on graphite-like carbon nitride. The work and the proposed approach to the synthesis are very interesting, modern methods of characterization have been applied and, in general, the whole study looks very worthy. I have only minor comments.

  1. I advise to move figures from Suppplementary to main text of the article.
  2. Actually, 3% Pt is a very high content of noble metal, I think the content of platinum can be decreased.
  3. The comparison of the activity with recently published data on g-C3N4 should be made.

Author Response

Detailed response on the comments by Reviewer #1

This manuscript is a very interesting work on the synthesis of a material based on graphite-like carbon nitride. The work and the proposed approach to the synthesis are very interesting, modern methods of characterization have been applied and, in general, the whole study looks very worthy. I have only minor comments.

Answer: Thanks for this positive comment. We have revised our manuscript to improve our paper quality according to the reviewer’s suggestion.

  1. I advise to move figures from Suppplementary to main text of the article.

Answer: Thanks for your kind advice. We have moved partial figures from Supplementary to main text of the article.

  1. Actually, 3% Pt is a very high content of noble metal, I think the content of platinum can be decreased.

Answer: Thanks for your question. We have referred to many related references (Appl. Catal. B-Environ. 2017, 204, 335–345; ACS Appl. Mater. Interfaces 2015, 7, 16850−16856; ACS Nano 2016, 10, 2745−2751), and, many of them used 3%wt Pt as co-catalyst in hydrogen evolution reaction, yet some references also used 1%wt Pt as co-catalyst. Hence, in the next work, we will try to reduce the content of Pt for research.

  1. The comparison of the activity with recently published data on g-C3N4 should be made.

Answer: Thanks for your suggestion. We have made a table to compare the published data on g-C3N4 of recent years.

Table S3.  Different g-C3N4 based materials for photocatalytic H2 evolution.

Materials

Co-catalysts

Light source

Sacrificial agent

H2 generation [µmol g−1 h−1]

AQE [%]

Stability at

least [h]

Ref.

2D-C3N4

Pd

>400nm

TEOA

1208.6

3.8 (420 nm)

20

[1]

g- C3N4

AgPd

>420nm

TEOA

900

-

-

[2]

g- C3N4

Au

780 nm >λ>420 nm

TEOA

350.6

1.5 (visible)

13

[3]

Oxygen doped g-C3N4

Pt

> 420 nm

TEOA

395.96

1.48 (500 nm)

20

[4]

g-C3N4(P)

Pt

≥ 420 nm

TEOA

916.2

6.52 (420 nm)

16

[5]

F-C3N4

Pt

>420 nm

TEOA

477.6

2.01 (420 nm)

25

[6]

B doped g-C3N4 quantum dots

Pt

>420 nm

TEOA

70.05 µmol h-1

10.0 (420 nm)

20

[7]

CNF

Pt

420-430 nm

-

1167.7

1.7 (420 nm)

25

[8]

g- C3N4

NiO

≥ 420 nm

TEOA

68.8

0.01 (500 nm)

30

[9]

g- C3N4

Ni2P

≥ 420 nm

TEOA

474.7

3.2 (435 nm)

20

[10]

F-g-CN

Pt

≥ 420 nm

TEOA

1298

4.53 (420 nm)

20

This work

References

[1] Mo Z.; Xu H.; She X.J.; Song Y.H.; Yan P.C.; Yi J.J.; Zhu X.W.; Lei Y.C.; Yuan S.Q.; Li H. Constructing Pd/2D-C3N4 composites for efficient photocatalytic H2 evolution through nonplasmon-induced bound electrons. Appl. Surf. Sci. 2019, 467, 151-157.

[2] Zou W.X.; Xu L.X.; Pu Y.; Cai H.J.; Wei X.Q.; Luo Y.D; Li L.L.; Gao B.; Wan H.Q.; Dong L. Advantageous interfacial effects of AgPd/g-C3N4 for photocatalytic hydrogen evolution: electronic structure and H2O dissociation. Chem. Eur. J. 2019, 25, 1-8.

[3] Tian H.Y.; Liu X.; Liang Z.Q.; Qiu P.Y.; Qian X.; Cui H.Z.; Tian J. Gold nanorods/g-C3N4 heterostructures for plasmon-enhanced photocatalytic H2 evolution in visible and near-infrared light. J. Colloid Interface Sci. 2019, 557, 700-708.

[4] Jiang Y.B.; Sun Z.Z.; Tang C.; Zhou Y.X.; Zeng L.; Huang L.M. Enhancement of photocatalytic hydrogen evolution activity of porous oxygen doped g-C3N4 with nitrogen defects induced by changing electron transition. Appl. Catal. B 2019, 240, 30-38.

[5] Wang B.; Cai H.R.; Zhao D.M.; Song, M.; Guo P.H.; Shen S.H.; Li D.S.; Yang S.C. Enhanced photocatalytic hydrogen evolution by partially replaced corner-site C atom with P in g-C3N4. Appl. Catal. B 2019, 244, 486-493.

[6] Ma F.K.; Sun C.L.; Shao Y.L.; Wu Y.Z.; Huang B.B.; Hao X.P. One-step exfoliation and fluorination of g-C3N4 nanosheets with enhanced photocatalytic activities. New J. Chem. 2017, 41, 3061.

[7] Wang Y.P.; Li J.L.; Zhao J.L.; Wang J.S.; Li Z.J. g-C3N4/B doped g-C3N4 quantum dots heterojunction photocatalysts for hydrogen evolution under visible light. Int. J. Hydrog. Energy 2019, 44. 618-628.

[8] Zeng L.; Ding X.; Sun Z.Z; Hua W.M.; Song W.L.; Liu S.Y.; Huang L.M. Enhancement of photocatalytic hydrogen evolution activity of g-C3N4 induced by structural distortion via post-fluorination treatment. Appl. Catal. B-Environ. 2018, 227, 276-284.

[9] Liu J.N.; Jia Q.H.; Long J.L.; Wang X.X.; Gao Z.W.; Gu Q. Amorphous NiO as co-catalyst for enhanced visible-light-driven hydrogen generation over g-C3N4 photocatalyst. Appl. Catal. B 2018, 222, 35-43.

[10] Zeng D.Q.; Xu W.J.; Ong W. J.; Xu J.; Ren H.; Chen Y.Z.; Zheng H.F.; Peng D.L. Toward noble-metal-free visible-light-driven photocatalytic hydrogen evolution: monodisperse sub–15 nm Ni2P nanoparticles anchored on porous g-C3N4 nanosheets to engineer 0D-2D heterojunction interfaces. Appl. Catal. B 2018, 221, 47-55.

Reviewer 2 Report

The manuscript reported by Li and Feng et al describes fluorination process of g-C3N4, thorough characterization, and photocatalytic activity in hydrogen evolution process. This manuscript is clearly written and it encompasses a new strategy to obtain such type of materials. I recommend the publication of this work in Nanomaterials after minor revisions and clarifications.

1) The writing of this manuscript needs to be improved to raise the quality of this paper. There are many typos (e.g., carries, alternative evacuation, imagines etc.) and inconsistent sentences (e.g., “Unlike conventional reaction with F-…”, line 73, “…the dense stacked lamellar 157 structure can be observed but it suppresses the capability of light absorption…”, lines 157-159).

2) The title of the article - “Fluorinated g-C3N4 with tunable defects and components for enhanced photocatalytic hydrogen evolution” is somewhat confusing. It sounds like F-g-CN besides tunable defects has components for enhanced photocatalytic hydrogen evolution. I recommend to propose new title.

2) First sentence of the second paragraph in Introduction section must be placed somewhere before the methods of the obtaining of fluorinated g-C3N4.

3) The authors state that the treatment with F-containing gas is widely applied method and provide only one reference (lines 60-63). In the same sentence, authors state, that such treatment is called the fluorination process. Are the treatment methods described earlier not fluorination processes?

4) There are no references on fluorination of graphite (line 65).

5) “…complete fluorination.” sounds too exaggerated and misleads the reader about the degree of fluoridation.

6) In the sentence on lines 159-161 authors mention term “fluorination” four times.

7) In the “Optical and electronic properties” section authors mention lifetime measurements but not provide the data.

8) According to Introduction section, the method of fluorination of g-C3N4 proposed by the authors allows to increase the doping level and modulate 56 the structure to improve the photocatalytic properties. Therefore, it is necessary to compare the results on photocatalytic hydrogen evolution efficiency with fluorinated g-C3N4 obtained by other methods known in literature (e.g., ref 22) or with carbon nitrides doped with other heteroatoms.

9) The authors used 3%wt Pt as co-catalyst in hydrogen evolution reaction. Pt was introduced as co-catalyst by in-situ photo-depositing of H2PtCl6. It is not clear, on what Pt nanoparticles are deposited? On g-C3N4/F-g-CN? If yes, how g-C3N4/F-g-CN coexist in solution – separately or as hybrid material? Does fluorinated state of F-g-CN affect the size of the particles and hence their quantity? Can this affect the efficiency of hydrogen evolution process? Is F-g-CN can act as catalyst by itself without co-catalyst?

Author Response

Detailed response on the comments by Reviewer #2

The manuscript reported by Li and Feng et al describes fluorination process of g-C3N4, thorough characterization, and photocatalytic activity in hydrogen evolution process. This manuscript is clearly written and it encompasses a new strategy to obtain such type of materials. I recommend the publication of this work in Nanomaterials after minor revisions and clarifications.

Answer: We thank the reviewer for the positive feed-back and the useful comments.

  1. The writing of this manuscript needs to be improved to raise the quality of this paper. There are many typos (e.g., carries, alternative evacuation, imagines etc.) and inconsistent sentences (e.g., “Unlike conventional reaction with F-…”, line 73, “…the dense stacked lamellar 157 structure can be observed but it suppresses the capability of light absorption…”, lines 157-159).

Answer: Thanks for your kind reminder. Mistakes and typos have been corrected in the main text by using red color. “Unlike conventional reaction with F-…”, this sentence describes the changes in the structure and properties of g-C3N4 after the introduction of fluorine atoms. While, “…the dense stacked lamellar structure can be observed but it suppresses the capability of light absorption…”, this sentence describes the morphology of g-C3N4. Therefore, there is no obvious inconsistency between the two sentences.

  1. The title of the article - “Fluorinated g-C3N4 with tunable defects and components for enhanced photocatalytic hydrogen evolution” is somewhat confusing. It sounds like F-g-CN besides tunable defects has components for enhanced photocatalytic hydrogen evolution. I recommend to propose new title.

Answer: Thanks for your very valuable suggestions. We have a new title which is “Gas-phase fluorination of g-C3N4 for enhance photocatalytic hydrogen evolution”.

  1. First sentence of the second paragraph in Introduction section must be placed somewhere before the methods of the obtaining of fluorinated g-C3N4.

Answer: Thanks for your very valuable suggestions. We put this sentence before the methods of the obtaining of fluorinated g-C3N4.

  1. The authors state that the treatment with F-containing gas is widely applied method and provide only one reference (lines 60-63). In the same sentence, authors state, that such treatment is called the fluorination process. Are the treatment methods described earlier not fluorination processes?

Answer: Thanks for your very valuable suggestions. We have added some literature reference on the treatment with F-containing gas. Since our description is not rigorous and caused unnecessary ambiguity, we have changed “fluorination processes” to “gas-phase fluorination processes”.

  1. There are no references on fluorination of graphite (line 65).

Answer: Thanks for your kind reminder. We have added a literature reference on fluorination of graphite.

  1. “…complete fluorination.” sounds too exaggerated and misleads the reader about the degree of fluoridation.

Answer: Thanks for your kind reminder. We have changed “complete fluorination” to “sufficient fluorination”.

  1. In the sentence on lines 159-161 authors mention term “fluorination” four times.

Answer: Thanks for your kind reminder. We have reorganized this section for clarity.

  1. In the “Optical and electronic properties” section authors mention lifetime measurements but not provide the data.

Answer: Thanks for your kind reminder. The time-resolved PL spectra were carried out at the wavelength of the maximum emission peak to measure the lifetime of the photogenerated charge carriers. The fluorescence decay curves were fitted by using the following equation: I(t) = A1exp (−t/τ1) + A2exp (−t/τ2) + A3exp (−t/τ3). In order to relate the overall emission decay behaviors, the average lifetime (τave) was evaluated using the following equation: τave = (A1τ12 + A2τ22+A3τ32)/(A1τ1 + A2τ2+A3τ3). It can be clearly deduced that after fluorination, the average emission lifetime of g-C3N4 and F-g-CN is 6.39 and 8.62 ns, respectively. The fluorescence lifetime of F-g-CN is longer than that of g-C3N4, suggesting that more charge carriers take part in the photocatalytic reaction.

  1. According to Introduction section, the method of fluorination of g-C3N4 proposed by the authors allows to increase the doping level and modulate 56 the structure to improve the photocatalytic properties. Therefore, it is necessary to compare the results on photocatalytic hydrogen evolution efficiency with fluorinated g-C3N4 obtained by other methods known in literature (e.g., ref 22) or with carbon nitrides doped with other heteroatoms.

Answer: Thanks for your suggestion. We have made a table to compare the published data on heteroatomic doping g-C3N4 and different co-catalyst of recent years.

Table S3.  Different g-C3N4 based materials for photocatalytic H2 evolution.

Materials

Co-catalysts

Light source

Sacrificial agent

H2 generation [µmol g−1 h−1]

AQE [%]

Stability at

least [h]

Ref.

2D-C3N4

Pd

>400nm

TEOA

1208.6

3.8 (420 nm)

20

[1]

g- C3N4

AgPd

>420nm

TEOA

900

-

-

[2]

g- C3N4

Au

780 nm >λ>420 nm

TEOA

350.6

1.5 (visible)

13

[3]

Oxygen doped g-C3N4

Pt

> 420 nm

TEOA

395.96

1.48 (500 nm)

20

[4]

g-C3N4(P)

Pt

≥ 420 nm

TEOA

916.2

6.52 (420 nm)

16

[5]

F-C3N4

Pt

>420 nm

TEOA

477.6

2.01 (420 nm)

25

[6]

B doped g-C3N4 quantum dots

Pt

>420 nm

TEOA

70.05 µmol h-1

10.0 (420 nm)

20

[7]

CNF

Pt

420-430 nm

-

1167.7

1.7 (420 nm)

25

[8]

g- C3N4

NiO

≥ 420 nm

TEOA

68.8

0.01 (500 nm)

30

[9]

g- C3N4

Ni2P

≥ 420 nm

TEOA

474.7

3.2 (435 nm)

20

[10]

F-g-CN

Pt

≥ 420 nm

TEOA

1298

4.53 (420 nm)

20

This work

Reference

[1] Mo Z.; Xu H.; She X.J.; Song Y.H.; Yan P.C.; Yi J.J.; Zhu X.W.; Lei Y.C.; Yuan S.Q.; Li H. Constructing Pd/2D-C3N4 composites for efficient photocatalytic H2 evolution through nonplasmon-induced bound electrons. Appl. Surf. Sci. 2019, 467, 151-157.

[2] Zou W.X.; Xu L.X.; Pu Y.; Cai H.J.; Wei X.Q.; Luo Y.D; Li L.L.; Gao B.; Wan H.Q.; Dong L. Advantageous interfacial effects of AgPd/g-C3N4 for photocatalytic hydrogen evolution: electronic structure and H2O dissociation. Chem. Eur. J. 2019, 25, 1-8.

[3] Tian H.Y.; Liu X.; Liang Z.Q.; Qiu P.Y.; Qian X.; Cui H.Z.; Tian J. Gold nanorods/g-C3N4 heterostructures for plasmon-enhanced photocatalytic H2 evolution in visible and near-infrared light. J. Colloid Interface Sci. 2019, 557, 700-708.

[4] Jiang Y.B.; Sun Z.Z.; Tang C.; Zhou Y.X.; Zeng L.; Huang L.M. Enhancement of photocatalytic hydrogen evolution activity of porous oxygen doped g-C3N4 with nitrogen defects induced by changing electron transition. Appl. Catal. B 2019, 240, 30-38.

[5] Wang B.; Cai H.R.; Zhao D.M.; Song, M.; Guo P.H.; Shen S.H.; Li D.S.; Yang S.C. Enhanced photocatalytic hydrogen evolution by partially replaced corner-site C atom with P in g-C3N4. Appl. Catal. B 2019, 244, 486-493.

[6] Ma F.K.; Sun C.L.; Shao Y.L.; Wu Y.Z.; Huang B.B.; Hao X.P. One-step exfoliation and fluorination of g-C3N4 nanosheets with enhanced photocatalytic activities. New J. Chem. 2017, 41, 3061.

[7] Wang Y.P.; Li J.L.; Zhao J.L.; Wang J.S.; Li Z.J. g-C3N4/B doped g-C3N4 quantum dots heterojunction photocatalysts for hydrogen evolution under visible light. Int. J. Hydrog. Energy 2019, 44. 618-628.

[8] Zeng L.; Ding X.; Sun Z.Z; Hua W.M.; Song W.L.; Liu S.Y.; Huang L.M. Enhancement of photocatalytic hydrogen evolution activity of g-C3N4 induced by structural distortion via post-fluorination treatment. Appl. Catal. B-Environ. 2018, 227, 276-284.

[9] Liu J.N.; Jia Q.H.; Long J.L.; Wang X.X.; Gao Z.W.; Gu Q. Amorphous NiO as co-catalyst for enhanced visible-light-driven hydrogen generation over g-C3N4 photocatalyst. Appl. Catal. B 2018, 222, 35-43.

[10] Zeng D.Q.; Xu W.J.; Ong W. J.; Xu J.; Ren H.; Chen Y.Z.; Zheng H.F.; Peng D.L. Toward noble-metal-free visible-light-driven photocatalytic hydrogen evolution: monodisperse sub–15 nm Ni2P nanoparticles anchored on porous g-C3N4 nanosheets to engineer 0D-2D heterojunction interfaces. Appl. Catal. B 2018, 221, 47-55.

  1. The authors used 3%wt Pt as co-catalyst in hydrogen evolution reaction. Pt was introduced as co-catalyst by in-situ photo-depositing of H2PtCl6. It is not clear, on what Pt nanoparticles are deposited? On g-C3N4/F-g-CN? If yes, how g-C3N4/F-g-CN coexist in solution – separately or as hybrid material? Does fluorinated state of F-g-CN affect the size of the particles and hence their quantity? Can this affect the efficiency of hydrogen evolution process? Is F-g-CN can act as catalyst by itself without co-catalyst?

Answer: Thanks for your question. Apologies. Since the photocatalytic measurement and the introduced 3 wt.% Pt as co-catalyst are carried out in professional testing institutions, we neglected to characterize the morphology of Pt/F-g-CN. Temporarily unable to provide the relevant morphologic characterization in a short time. As well as, the morphology characterization of g-C3N4/co-catalyst (Pt) composites have not been described in many relative references (Such as ACS Appl. Mater. Interfaces 2015, 7, 16850−16856; Appl. Surf. Sci. 2019, 495, 143555). F-g-CN and g-C3N4 can act as catalyst by itself but hydrogen evolution rate is very low (J. Mater. Chem. A 2018, 6, 13110-13122; Appl. Catal. B-Environ. 2018, 220, 379–385; ACS Appl. Mater. Interfaces 2019, 11, 5651). The recently reported about fluorinated g-C3N4 are all carried out with the participation of co-catalysts (Pt) (Such as Appl. Surf. Sci. 2019, 474, 194-202; New J. Chem., 2017, 41, 3061; Appl. Catal. B-Environ. 2018, 227, 276-284). The HER activity of modified g-C3N4 is limited by insufficient surface catalytic sites, despite of its suitable bandgap for light absorption and chemical stability. Actually, cocatalysts are indispensable and of pivotal importance in the third process for g-C3N4 HER system. Hence, the separation of photogenerated charge carriers and utilization of solar energy is improved by the N vacancy defects, as well as by the presence of introduced F atoms and the structural distortion induced by fluorination. It also required the utilization of cocatalysts in the g-C3N4-based photocatalytic systems due to the positive effects of cocatalysts on supressing charge carrier recombination, reducing the HER overpotential, and improving photocatalytic activity.

Reviewer 3 Report

The manuscript deals with the formation of vacancies in graphitic carbon nitride using fluorine in a mixture with nitrogen. The manuscript is interesting but there are results of one sample only. I think the results should be repeated for the different contents of fluorine in the mixtures. Therefore, I recommend major revision in which new data will be collected and presented.

Author Response

Detailed response on the comments by Reviewer #3

The manuscript deals with the formation of vacancies in graphitic carbon nitride using fluorine in a mixture with nitrogen. The manuscript is interesting but there are results of one sample only. I think the results should be repeated for the different contents of fluorine in the mixtures. Therefore, I recommend major revision in which new data will be collected and presented.

Answer: We thank the reviewer for the positive feed-back and the useful comments. We have added the surface atom content of different fluorinated temperature g-C3N4 (Table S2) as well as transient photocurrent responses and time dependent photocatalytic H2 evolution of g-C3N4 and F-g-CN with different fluorinated temperature (Figure S7). The F concentration increased with the increase of fluorination temperature, and the porous structure of F-g-CN obtained at 180 ËšC is seriously deteriorated due to the extensive fluorination at a high temperature. Additionally, the photocurrent density of g-C3N4, F-g-CN-120, F-g-CN-150, F-g-CN-180 are 3.54, 4.58, 8.82 and 4.28 μAcm-2 (Figure S7a), respectively, indicating that F-g-CN-180 can provide more photogenerated charge carriers. As shown in Figure S7b, the average hydrogen evolution rate of g-C3N4, F-g-CN-120, F-g-CN-150 and F-g-CN-180 are about 112, 675, 1298, 370 μmol h-1 g-1. It is clearly to see that the hydrogen evolution rate of F-g-CN-150 is much higher than others. The etched bridged N atoms during fluorination generate large number of nitrogen defects as vacancies and C-F bonds on the edge of tri-s-triazine, which introduce defect state and F dopant level into the electronic band of F-g-CN. The g-C3N4 obtains relatively insufficient nitrogen defects as vacancies and C-F bonds at lower fluorinated temperatures, while seriously deteriorated nitrogen defects at higher fluorinated temperatures. Appropriate defects and fluorine doping can achieve excellent performance.

Figure S7. (a) Transient photocurrent responses of g-C3N4 and F-g-CN with different fluorinated temperature. (b) Time dependent photocatalytic H2 evolution over the pristine and the fluorinated g-C3N4 samples.

Table S2. The surface atom content of different materials

C (at. %)

N (at. %)

O (at. %)

F (at. %)

g-C3N4

52.10

45.13

2.77

0

F-g-CN-120

46.80

45.24

3.12

3.69

F-g-CN-150

46.31

43.21

3.43

7.05

F-g-CN-180

44.65

39.5

3.93

9.37

Round 2

Reviewer 3 Report

The manuscript was improved and can be published.